# Are racial and gender inequalities emerging in the NFT artwork? A visual exploration of CryptoPunks

Yufan Zhang [*]
Duke Kunshan University
SciEcon CIC

Zichao Chen[†]
Duke Kunshan University
SciEcon CIC

Chang Xu
Hunan University

Luyao Zhang
Data Science Research
Center
Duke Kunshan University
SciEcon CIC

Xin Tong[‡]
Data Science Research Center
Division of Arts and Humanities
Duke Kunshan University

## ABSTRACT

As a blockchain-based application, Non-Fungible Token (NFT) has received worldwide attention over the past few years. Digital artwork is the main form of NFT that can be stored on different blockchains. Although the NFT market is rapidly developing, potential ethical and racial fairness issues are arising in NFT artworks due to a lack of ethical guidelines or censorship. Therefore, in this research, we investigated the potential ethical issues of CryptoPunk, one of the most long-lasting NFT collections, and visualized its visual appearance's genetic distribution, associations between selling prices, gender/skin tone factors, and transaction network. We further explored the ethical issues related to CryptoPunks from three aspects: design, trading transactions, and related topics on Twitter. We provided five visualization sections to highlight our findings about CryptoPunks' ethics issues. From our data analysis and visualization interpretations, we found that (1) around 1.6 times more male punks were created in the initial design process than the female ones; (2) lighter-skinned punks tend to sell for higher prices; (3) the topic related to identity equality in NFTs is rarely mentioned on Twitter. These findings and visualizations from CryptoPunks example provide a preliminary understanding of the racial and gender inequity ethical problems, and also a potential approach for exploring future ethical-related research in the NFT domain.

## 1 INTRODUCTION

As blockchain technology continues to be popular around the world, its core features, immutable, security, and transparency are entering the public's mind [1]. With these exciting features, blockchain technology is driving a lot of revolutions in plenty of areas and one of the hottest and fast-growing applications is Non-fungible token (NFT). NFT is a unit of data stored on the blockchain that certifies a digital asset to be unique [2]. More and more developers, artists, and collectors are entering the NFT art community [1]. The NFT market achieved more than 2 billion dollars in transactions in the first quarter of 2021 — which is 2000% higher than the fourth quarter of 2020 [3]. Besides, many famous artists and companies have also joined the NFT market to redefine the way of digital artwork [4, 5]. Numerous NFT digital avatar collections are created based on attributes like the avatar's gender, skin color, and accessories, of which CryptoPunk [6] is one of the most popular NFT collections that offers more

---

[*]The first two authors are co-first authors.

[†]Co-first authors

[‡]Communication authors: Xin Tong xt43@duke.edu and Luyao Zhang lz183@duke.edu

than 10,000 unique collectible character images, selling on every mainstream NFT marketplace. However, without regulation, ethical problems have the risk to be out of control in the fledgling NFT market. For example, a writer from Bloomberg has taken note of the potential for ethical inequality in CryptoPunks, where punks with different genders and skin colors may have different price trends [7]. The author speculates from recent price trends that CryptoPunk with light skin and male gender could command higher prices. Moreover, the algorithmic or manual generation processes of these digital artwork collections usually remain unrevealed. The NFT transaction process has been proven to be embedded with discriminatory risks [8]. Stakeholders getting involved in the NFT generation and transaction process can all cause potential discrimination risks. For instance, Kizhner, et al [9] found that digital content and artworks may carry cultural biases in the generation and selection stages. Similarly, other members from crypto online communities can also carry their subjective perceptions, interpretations, wishes, or emotions to appearances when trading NFT artwork or sharing information about them via social media [10]. Therefore, we aim to answer the following research questions through data visualizations:

- What potential gender and racial ethical concerns exist in NFT artwork collections in the generation and trading process, such as CryptoPunks?

- What are Twitter users' perceptions of NFT ethical problems?

In the result, our analysis and visualizations revealed that inequality appears in both the generating and trading process. In the generating period, the number of male CryptoPunks is 1.6 times higher than that of female CryptoPunks. And in the trading record since 2022, the median prices of CryptoPunks with light, medium, and dark skin are 44.72 ETH, 43.85 ETH, and 41.99 ETH respectively. It shows that the price of light-skinned punks is obviously higher than that of dark-skinned punks. Besides, we also analyzed the transaction network, and we found that there were a lot of big merchants dominating the trading network. Since the launching of the CryptoPunks NFT project, 65% of the CryptoPunks transactions took place in merely 10% of the addresses that appeared in the transaction network. From Twitter data, we found that the current discussions on NFT ethics did not involve much information about gender and ethical equality, which indicated that people are not aware of the potential problems. Our innovative visualization and data analysis methods bring the inequality of CryptoPunk to the audience in a more intuitive way. As one of the first groups to explore this question scientifically, our result provides an initial ethics inequality exploration of an NFT collection case that could further enlighten ethics-related NFT investigations and research.

## 2 METHODOLOGY

### 2.1 Data Collection

The data were scraped from two sources: blockchain transaction records of CryptoPunk token from DuneAnalytics [11] ranging from the launch date of CryptoPunk, June 23, 2017, to July 27, 2022, and the Twitter threads with at least 5 likes from Twitter [12] with related topic keywords. The rule is that the tweets must include one word in "NFT" or "CryptoPunk", and also include any word in "ethic", "informed consent", "transparency", "accountability", "privacy", "fairness", "trust", "gender", "ethnicity", "skin tone", and "skin color".

### 2.2 Data Processing and Filtering

#### 2.2.1 Construct the databases

Using the Python Pandas [13] and Numpy [14] libraries, we construct four databases: transaction database, token database, trader database, and tweet database, whose primary keys are the transaction ID, CryptoPunk ID, Ethereum address, and Twitter thread ID respectively. Specifically, 16,823 CryptoPunk transactions are recorded in the transaction database, where each item contains the timestamp, the addresses that sold and received the token, the transaction value in ETH, and the token ID referring to the CryptoPunk. The token database records the 10,000 unique CryptoPunk characters, each of which contains the attributes of the CryptoPunk, including its gender, race, skin tone, and other attributes, and its average price. A total of 5,911 addresses that involved at least one CryptoPunk transaction are recorded in the trader database, where each item contains all the token IDs that this address used to own and owns now; and 83,568 Twitter threads are recorded in the tweet database, containing the timestamp and content of the tweets.

#### 2.2.2 Extract data for visualization

To depict the distribution of CryptoPunk with different attributes, we group the CryptoPunk tokens with their attributes. Specifically, the attributes of CryptoPunk are categorized into four levels: the type (e.g., human), the gender (e.g., male), the skin tone (e.g., dark), and the number of attributes (e.g., 3 attributes). Furthermore, the transaction database is merged with the token database to depict the differences in price for CryptPunk based on ethics-related attributes such as gender and skin tone, as well as how the differences change over time. Using the Python NetworkX [15] library, we build the CryptoPunk transaction network for each year from 2017 to 2022 with the addresses as the nodes and the transaction as the edges.

#### 2.2.3 Conduct sentiment analysis

Using the Python flairNLP [16] library, we conducted sentiment analysis on the tweet content in the tweet database. Unlike the rule-based sentiment analysis toolkits such as Textblob [17], flairNLP assigns each text with a latent embedding via the deep neural network, which generally yields better model performance in terms of Natural Language Processing (NLP) tasks such as sentiment analysis. Specifically, each tweet is classified into positive, neutral, or negative sentiment: 47,831 (57.46%) tweets are predicted as positive sentiment; 23,567 (28.31%) tweets are predicted as neutral sentiment; 11,847 (14.23%) tweets are predicted as negative sentiment.

### 2.3 Data Visualization Method

Using the Python Plotly [18] library, we construct five visualization figures (Figures 1-8) based on the processed data. First, we explore the distribution of CryptoPunk with different attributes by creating an interactive Sankey diagram. Then, we apply a series design idiom to depict the price disparity based on skin tone. Based on the undirected graph of the transaction network by year, we create a circular network visualization where each edge between nodes is colored with the corresponding CryptoPunk skin tone. For the analysis of the transaction network, we also draw the dynamic graph of the network features. Moreover, we create the bar charts and word cloud using Python WordCloud [19] package to visualize Twitter users' sentiment towards NFT-related ethical topics. See more details in the visualization results section.

## 3 RESULTS

Based on the datasets we got in section 2.2, we designed five interactive visualizations that allow users to intuitively view the potential inequities that exist in CryptoPunks.

### 3.1 CryptoPunks Generating Distribution

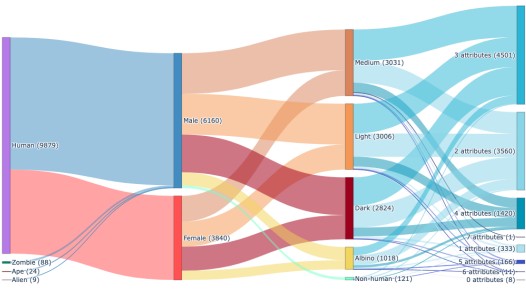

Figure 1: Sankey Diagram of the Attributes Distribution

Figure 1 illustrates the differences in the CryptoPunks genesis with various attributes where 6160 (61.6%) of the punks are with male type and 3840 (38.4%) of the punks are with the female type. Among the 9879 human CryptoPunks, 3031 (30.68%) of the punks with a medium skin tone, 3006 (30.42%) of the punks with a light skin tone, 2824 (28.59%) of the punks with a dark skin tone, and 1018 (10.30%) of the punks with the albino skin tone. Looking at the generation phase, we saw a big difference in the number of CryptoPunks for males and females. And the number varies slightly by skin color. We searched for documentation about the original design of CryptoPunk, but couldn't find a description of the initial Punk quantity allocation ratio. From the current research, we can't conclude if this is a reasonable allocation, but there is indeed a number inequality between male and female features.

### 3.2 Price Differences of CryptoPunks with Different Skin Tones

Among the 87 attributes (e.g., Top Hat, Gold Chain, 3D Glasses) in total in the CryptoPunks NFT collection, we observed that a large percentage of the cheapest punks in selling lists currently under each attribute are with darker skin tone, which is illustrated in Figure 2. Specifically, there are 41 attributes (e.g., Chinstrap, Mustache) whose cheapest punks are with dark skin tone, 25 attributes (e.g., Eye Mask, Bandana) whose cheapest punks are with medium skin tone, 16 attributes (e.g., Cigarette, Cowboy Hat) whose cheapest punks are with light skin tone, and 5 attributes (e.g., Red Mohawk, Handlebars) whose cheapest punks are with albino skin tone. The number above means that when we observe the price of CryptoPunks in the dimension of the attribute, we can find that the cheapest Punk in multiple attribute sets is the ones with dark skin, while CryptoPunk with light skin color rarely appears in the cheapest position.

To further depict the price differences for CryptoPunks with different skin tones, we choose the transaction records starting from Jan 1th 2022, to visualize the price distribution for each skin tone. Moreover, we explore the price distribution of skin tones of punks with different numbers of attributes (i.e., 1, 2, 3, 4, 5), which is illustrated in Figure 3.

Specifically, the median prices are 41.99 ETH for the punks with dark skin tone, 43.85 ETH for the punks with medium skin

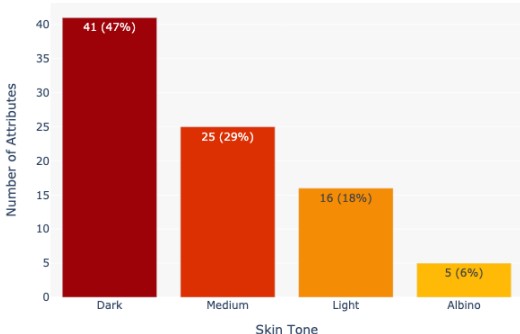

Figure 2: Distribution of the skin tone of the cheapest punk for each attribute

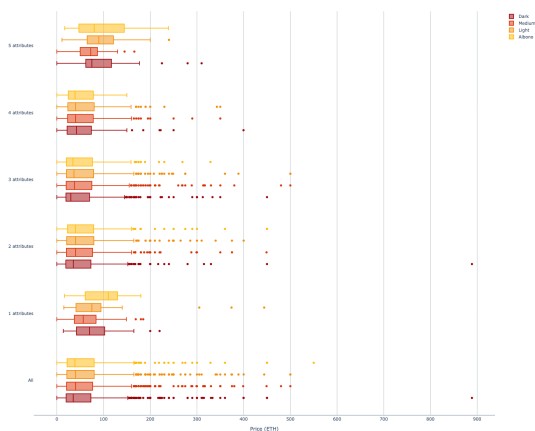

Figure 3: Distribution of the CryptoPunk price of different skin tones (after 2021)

tone, 44.72 ETH for the punks with light skin tone, and 40 ETH for the punks with albino skin tone. Since Albino is a rare Punk attribute (9 in 10000), we hypothesize that the low median price is probably because of their low number of cases in the trade. With the exception of Albino Skin Tone, for the other three primary Skin tones, we found that CryptoPunk with darker colors had a lower median price in the transaction.

### 3.3 Price Difference Trends over Time

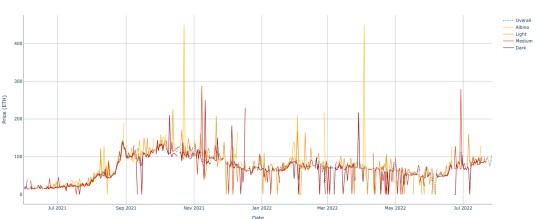

Figure 4: Time series plot of the median prices of CryptoPunks with different skin tones (after May 31, 2020)

We further explore how the price difference among punks with different skin tones changes over time, especially since the CryptoPunks project began to draw more and more attention from the world in 2021. Figure 4 shows the median prices of CryptoPunks with different skin tones after May 31, 2020. We can observe that the overall price trends for punks with various skin tones are very similar; that is, they all follow the overall price trend for the whole CryptoPunk collection. Nevertheless, Figure 4 also indicates that most of the price gap between punks with lighter skin tones and darker skin tones does exist, especially where the most expensive CryptoPunk transactions are often associated with punks with lighter skin tones. Furthermore, we illustrate the CryptoPunks transaction in regard to the punks' skin tones using a scatter plot (Figure 5), where each scatters denotes a CryptoPunk transaction whose color represents the punk's skin tone. Similar to Figure 4, we can observe that most transactions associated with punks with darker skin tones generally show lower transaction prices compared to other transactions on the same date.

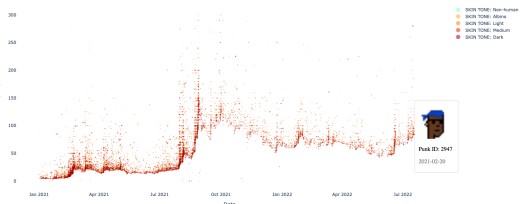

Figure 5: CryptoPunks transactions in terms of the punk's skin tone (after Dec 31, 2020)

### 3.4 CryptoPunk Transaction Network Analysis

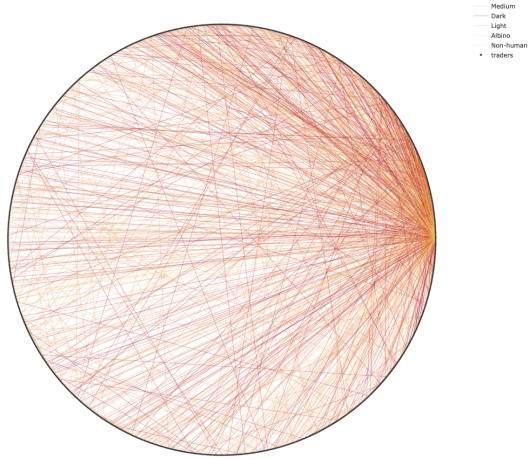

Figure 6: Circular network of the CryptoPunks transactions in 2022

Besides visualizing the price changes and distributions of each kind of CryptoPunks, we are eager to figure out the reason behind this situation. Since the trading prices in the market are set by traders, we took an in-depth look at CryptoPunk's trading network. We explore the potential causes for the price difference with different skin tones from the perspective of the transition network. In Figure 6, each node refers to a trader who has participated in at least one transaction of CryptoPunk in 2022, and each edge refers to a transaction with the color representing the punk skin tone of the transaction. In this network graph, the right side of the transaction circle has the densest edges, which means that prices in the CryptoPunk market are likely to be dominated by a few big business merchants or high-frequency collectors. In fact, since the launching of the CryptoPunks NFT project, 65% (23159 out of 35652) of the CryptoPunks transactions took place in merely 10% (643 out of 6438) of the addresses that have ever participated in at least one CryptoPunks transaction.

In other words, the trading market of CryptoPunks with approximately $2.9B volume is controlled by a small number of addresses. Moreover, investigate the detailed account information of the 6 most active participants of CryptoPunks by Etherscan.io, detailed in Table 1.

| address | # of tx | who |
|---------|---------|-----|
| x1919db36ca2fa2e15f9000fd9cdc2edcf863e685 | 997 | Punks OTC |
| x53ede7cae3eb6a7d11429fe589c0278c9acbe21a | 768 | Rarible: Braithwaite |
| xd387a6e4e84a6c86bd90c158c6028a58cc8ac459 | 600 | Pranksy |
| x00d7c902fbbcd3c9db2da80a439c94486c50eb81 | 396 | Unknown |
| x269616d549d7e8eaa82dfb17028d0b212d11232a | 371 | BeaconProxy (Contract) |
| x577ebc5de943e35cdf9ecb5bbe1f7d7cb6c7c647 | 303 | Unknown |

Table 1: Most active participants in the CryptoPunks market

Following [20], we then use the Blockchain Network Analysis to further examine how the centralization of the CryptoPunks changes over time and whether its future trend is toward decentralization, which is the basis of the egalitarian ideals that blockchain technology has promised to provide. A more decentralized transaction network should show a higher number of components and modularity, a lower quantity of the grain component size ratio, and the stand deviation of the degree centrality [20]. As illustrated in Figure 7, the network features dynamics of the CryptoPunks transaction network after Dec 31, 2020, indicates that the CryptoPunks transaction network first becomes decentralized at around the start of 2021 and then becomes centralized after around Oct 2021.

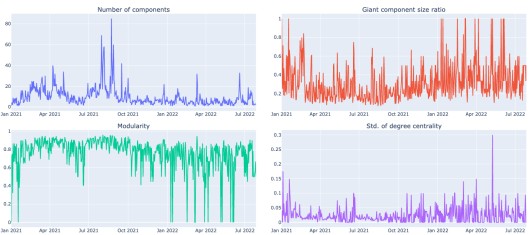

Figure 7: Network feature dynamics of the CryptoPunks transaction network

### 3.5 Twitter Analysis with Bar Chart Word Cloud

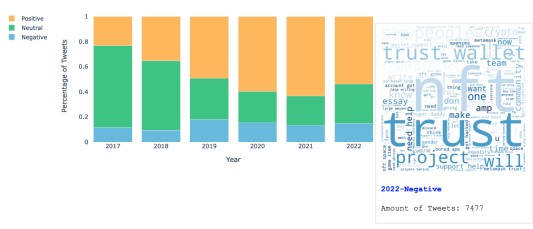

Figure 8: Bar Chart of the Word Cloud and Sentiment Analysis

The bar chart and the word cloud visualizations in Figure 5 reflected the sentiments of Twitter users' attitudes about NFT ethics. The keywords we chose included NFT, ethics, fairness, gender, race, equality, and so on. By clicking on each section, we can see the main discussion word cloud under this section on the right. Even though the NFT trust topic is also important [21], we believe gender and racial inequities are also an essential part of NFT ethics. However, we found that very few users were tweeting about race and gender

equality in NFT topics. Combining the data analysis and conclusions of the previous four sections with the Twitter word cloud we present, we find that gender and racial inequalities do exist in the generation and transaction process, but Twitter users are not talking about these things. This means that the problem is existing, but the attention is missing.

## 4 DISCUSSIONS AND CONCLUSION

NFT is a very hot and popular market right now, and the number of players is growing rapidly. However, as we discussed in the introduction section, the NFT market currently presents potential ethical and fairness issues. In this project, we successfully collected and processed the data about generating and the transaction network from CryptoPunk and NFT ethics-related Twitter discussions. We created a series of highly interactive visualizations to demonstrate the potential equity issues in a more intuitive way, and our interactive visualizations will be published online for broader users. Our visualizations showed that different levels of gender and racial inequality have emerged both during the creation (as shown in Figure 1) and trading stages of Cryptopunk (as shown in Figures 2-7). In the CryptoPunk generating stage, the number of different genders and races of Cryptopunks are unevenly distributed, where the amount of male CryptoPunks is 1.6 times higher than female CryptoPunks. Then in the trading stage, our result reveals that the market price varies according to race. The CryptoPunk with lighter skin color will lead to a higher selling price. We also applied network analysis to the transaction network and found that large merchants (about 10% of all traders) dominate around 65% of transactions in the trading network, which may be the determinant of the market price. In the end, our paper also examines the discussion of inequality in NFTs among users on social platform Twitter. However, our findings suggest that while inequality indeed exists in both the generation and trading processes, it is rarely noticed by the public.

As an exploratory and innovative research article, this paper has some inspirations for future research. First of all, we have not carried out enough quantitative analysis. In the next step, we can introduce some mathematical models to analyze the degree of inequality in CryptoPunks. Second, we can conduct more in-depth research and interviews in the future. As the generating algorithm of Cryptopunks is still a black box, whether the initial distribution inequality is due to bias from artists or algorithmic mechanisms still needs investigation.

Not only should we focus on gender and racial equality issues, but more broadly, we should focus on the diversity and inclusion issues in NFT. Or how disadvantaged groups are protected for opportunity equality to be included and prosper in the NFT market places [22]. As researchers, developers, and customers working on blockchain, we need to jointly focus on enabling the NFT art community to implement the goals of blockchain, which is to achieve true equality, freedom, and inclusiveness.

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
