# OpenReview forum: "Are racial and gender inequalities emerging in the NFT artwork? A visual exploration of CryptoPunks"
_IEEE.org/2022/Workshop/altVIS — Reject_

### Official Review · Reviewer_ofdG · 2022-08-09

**Review:**

I feel this contribution doesn't fit the purpose of alt.VIS. It doesn't comment on or advances the visualization field in any way.
The paper proposes potential issues that are arising from the dataset. It would be worth mentioning that the problems arise mainly from the actual demographic trading the NFTs rather than from the trade itself. It specifies a research question: "What are Twitter users' perceptions of NFT ethical problems?" But why is Twitter important for the discussion? It isn't apparent from the text.
The claim that people are unaware of the potential problems mentioned in the introduction is bold. Could it be that perhaps they simply don't care?
The visualizations are of rather poor quality and hard to read and understand. I don't recommend this publication.

**Conflicts:**

None.

**Review Inclusion:**

No

**Sufficiently Alt:**

No

**Superlative:**

The most racially biased.

---

### Official Review · Reviewer_eR4s · 2022-08-09

**Review:**

This paper reads like someone pulled trending topics from twitter and decided to write a paper on them.
Authors can't really state that there's a significant difference in the NFT prices by only stating the median - including the variance and possibly a p-value analysis would have made the argument much stronger. Figure 3 shows a chart with some values, but a reader can't infer the values from that. The same figure starts from 2021, but the choice of limiting the data from 2021 is not justified.

The claim about the price of albino cryptopunks ("Since Albino is a rare Punk attribute (9 in 10000), we hypothesize that the low median price is probably because of their low number of cases in the trade") sounds weird - why would scarcity lead to lower prices? It makes me wonder why did the authors decide to use the median instead of the mean.

I believe the choice of words used in filtering tweets is limited and it would need some more discussion and justification too.
Would all tweets regarding skin tone in cryptopunks mention the words “ethnicity”, “skin tone”, or “skin color”? Because that feels like a limited set of words. I don't necessarily want to suggest that that list of words might be expanded, rather I'd like to say that this type of analysis feels a little inconclusive because of how hard it is to get a fair set of data points.

Overall, I believe the claims in this paper do not have much value.

**Conflicts:**

I have no conflicts with the authors

**Review Inclusion:**

No

**Sufficiently Alt:**

No

**Superlative:**

Twitterest

---

### Official Review · Reviewer_BMqH · 2022-08-30

**Review:**

Summary review: The paper purports to be a “visual exploration” of the NFT trading data for a popular digital asset, CryptoPunks, with specific focus on the demographics of the characters traded. However, this paper reads more as a loose statistical analysis motivated by vaguely defined ‘ethics’ with some standard visualization supplemental material. While this is a potentially important area of research, it seems like a better fit for a conference on data ethics, with significant improvements to the hypothesis testing process.

There is no discussion in the introduction about the well-documented potential market manipulation in NFTs such as wash trading.

Choices are not supported by literature; for example, why are the keywords in 2.1 selected, and what basis exists in the literature for selecting these keywords? Why did the authors choose to make the transaction graph undirected (or was this simply a limit of their choice to use Plotly for visualization)?

Bibliography is severely lacking and consists mostly of software packages. Authors have not meaningfully engaged in literature related to cryptocurrencies, digital art, data ethics, or data visualization. There are no citations in the majority of the paper.

“our findings suggest that while inequality indeed exists in both the generation and trading processes, it is rarely noticed by the public” this is inconsistent with the mechanic of ‘rarity’ which plays a major role in determining price and is frequently discussed among traders.

This paper needs significant grammatical revision.

Visualizations presented violate some of the most basic principles of data visualization creation, such as readability. The word cloud is particularly egregious, and even shows some tokenization errors, as the word “trust” appears twice.

“Figure 4 also indicates that most of the price gap between punks with lighter skin tones and darker skin tones does exist” this conclusion is not apparent from Figure 4. In general, many of these assertions should be supported with statistical analysis and the inclusion of p-values, t-tests, etc. There is no hypothesis testing presented here; nor is ‘holistic evaluation from visualization’ presented as a provocation for the paper. “the price of light-skinned punks is obviously higher than that of dark- skinned punks” cannot be determined by simply comparing medians.

This is not a data visualization paper; it simply uses two data visualization packages to visualize data, but does not meaningfully engage with or contribute to research on data visualization. alt.VIS is a workshop at a conference on data visualization, and therefore this paper is not a good fit.

**Conflicts:**

None

**Review Inclusion:**

Yes

**Sufficiently Alt:**

No

---

### Official Review · Reviewer_siDE · 2022-08-31

**Review:**

Meta-review:
----------------

Overall, the reviews of this submission have indicated that despite the interest it could yield, it is not currently in a shape that would make it suitable for the alt.VIS workshop. The main point that arises from the reviews is that the paper is not, in itself, "weird" enough to be an alt.VIS contribution, by its format or topic. In addition, the reviewers of the submission have highlighted limitations that should probably be addressed by the authors before they try and submit this work to a more fitting venue.

**Conflicts:**

No conflict of interests to declare.

**Review Inclusion:**

Yes

**Sufficiently Alt:**

No

---

### Decision · Program_Chairs · 2022-08-31

Reject